# Enhanced Antibiotic Tolerance of an *In Vitro* Multispecies Uropathogen Biofilm Model, Useful for Studies of Catheter-Associated Urinary Tract Infections

**DOI:** 10.3390/microorganisms10061207

**Published:** 2022-06-13

**Authors:** Jiapeng Hou, Lutian Wang, Martin Alm, Peter Thomsen, Tor Monsen, Madeleine Ramstedt, Mette Burmølle

**Affiliations:** 1Section of Microbiology, Department of Biology, University of Copenhagen, Universitetsparken 15, 2100 Copenhagen, Denmark; jiapeng.hou@bio.ku.dk (J.H.); lutianwang@bio.ku.dk (L.W.); 2Biomodics ApS, Fjeldhammervej 15, 2610 Rødovre, Denmark; ma@biomodics.com (M.A.); pt@biomodics.com (P.T.); 3Department of Clinical Microbiology, Umeå University, 901 85 Umeå, Sweden; tor.monsen@umu.se; 4Department of Chemistry, Umeå University, 901 87 Umeå, Sweden; madeleine.ramstedt@umu.se

**Keywords:** biofilms, CAUTI, multispecies, infections, interactions, antibiotic tolerance

## Abstract

Catheter-associated urinary tract infections (CAUTI) are a common clinical concern as they can lead to severe, persistent infections or bacteremia in long-term catheterized patients. This type of CAUTI is difficult to eradicate, as they are caused by multispecies biofilms that may have reduced susceptibility to antibiotics. Many new strategies to tackle CAUTI have been proposed in the past decade, including antibiotic combination treatments, surface modification and probiotic usage. However, those strategies were mainly assessed on mono- or dual-species biofilms that hardly represent the long-term CAUTI cases where, normally, 2–4 or even more species can be involved. We developed a four-species *in vitro* biofilm model on catheters involving clinical strains of *Escherichia coli*, *Pseudomonas aeruginosa*, *Klebsiella oxytoca* and *Proteus mirabilis* isolated from indwelling catheters. Interspecies interactions and responses to antibiotics were quantitatively assessed. Collaborative as well as competitive interactions were found among members in our model biofilm and those interactions affected the individual species’ abundances upon exposure to antibiotics as mono-, dual- or multispecies biofilms. Our study shows complex interactions between species during the assessment of CAUTI control strategies for biofilms and highlights the necessity of evaluating treatment and control regimes in a multispecies setting.

## 1. Introduction

Infections can start to develop immediately after the insertion of catheters and the risk of getting infections increases with the duration of catheterization [1,2]. The presence of bacteria in urine, bacteriuria, is generally observed in 100% of all patients catheterized for a time period of more than 2–4 weeks [3]. As long as no symptoms start to develop, patients are diagnosed as having asymptomatic bacteriuria where no antibiotics are prescribed [4]. These bacteria can develop biofilms with reduced susceptibility to, e.g., antibiotics or fluid flush. Several factors influence biofilm formation, for example, some bacteria may have higher probability to settle on catheter surfaces due to limited fluid intake, improper bag positioning or insufficient bag drainage [5,6]. As the period of catheterization becomes longer, the biofilm will continue developing and patients will start suffering from infection symptoms, e.g., pain, fever, dysuria, etc. [7], which is diagnosed as symptomatic UTI. There is a high risk of symptomatic UTI and catheter blockage due to the formation of biofilms for patients with long-term (>30 days) catheterization [8]. An effective way to eliminate or prevent infections is to replace the catheter. However, there is still a high rate of infection recurrence after catheter replacement, especially for women with an estimated risk of almost 50% to experience at least one recurrent infection within a year [9]. The consequences of the recurrent infections include patient suffering and inability to work, as well as an increased burden on hospital resources [5,10].

Many strategies have been proposed and tested to reduce viability, bacterial adhesion or the biofilm growth of uropathogens, including the use of antibiotics, antimicrobials, probiotics, phages, hydrophilic or antimicrobial surface coatings, and innovative catheter designs [11,12]. None of these “solutions” reduce the clinical risk of infections [13]. Part of the reason might be that these strategies were mostly evaluated on single strains or species. However, most of the clinical bacteriuria and UTI cases occurring in long-term catheterized patients are polymicrobial [14,15]. In hospitalized CAUTI patients, many catheters contain combinations of 2–4 strains each, and even cases with up to 6 strains have been reported [16]. The possible interspecies interactions taking place between these strains may induce antibiotic resistance by, e.g., horizontal gene transfer, enhancement of antibiotic tolerance of the bacteria present, promotion of biofilm formation, or alteration of secreted metabolites that inactivate antimicrobials [14,17]. For example, strains resistant against nine commonly prescribed antibiotics for UTI from the bacterial species of *Escherichia coli*, *Enterococcus faecalis*, *Staphylococcus aureus* and *Pseudomonas aeruginosa* were found to be more prevalent in polymicrobial compared to monomicrobial infection cases [18]. In addition, *S. aureus* gained 10–100-fold higher tolerance to 5 antibiotics when co-localized with *Candida albicans* [19]. Therefore, the development of polymicrobial models for the evaluation of UTI control strategies is urgently needed.

According to clinical and healthcare records of the past 20 years, the most frequent pathogens involved in CAUTIs are *E. coli*, *P. aeruginosa*, *Klebsiella* spp., *Enterococcus* spp., and *Proteus mirabilis* [4,10,20]. Among them, *E. coli*, *P. aeruginosa*, *Klebsiella* spp. and *Enterococcus* spp. were typically found as early colonizers, especially in patients with short-term catheterization, while the more challenging species including *P. aeruginosa*, *P. mirabilis* and *Morganella morganii* are more frequently involved in long-term indwelling catheters (>28 days) [21,22]. *P. mirabilis* is well known to cause the most severe problems due to its ability to produce potent urease, increasing the urine pH and causing the precipitation of salts that may lead to encrustation. Mature biofilms lead to many problems, including catheter blockage, pyelonephritis, septicemia as well as stone formation in the bladder and kidney [23]. In addition to causing problems by itself, *P. mirabilis* provides benefits for other strains by promoting their adhesion and sheltering them in the crystalline biofilm [8]. The swarming ability of *P. mirabilis* also facilitates the mobility of other non-motile species such as *Klebsiella pneumoniae* and *S. aureus* on the catheter surface [22].

Pairwise interactions between commonly isolated CAUTI species have been previously studied. These studies have shown that interspecies competition as well as cooperation altered species abundance [24], virulence factor production [18], metabolic activity [25,26], biofilm formation [24], antimicrobial tolerance [27] and antibiotic resistance [28]. Clinical evidence has shown the possibility of the co-localization of more than two species at the infection site [16]. Furthermore, the difference in species prevalence between initial colonizers and biofilm in long-term catheterized patients indicates that a dynamic polymicrobial community is formed [14] where multiple species are interacting. During colonization, interactions between the two original species could be changed when a third or more species joins the community [29].

In this study, we aimed to develop and characterize an *in vitro* four-species urinary catheter biofilm model using clinical isolates from indwelling catheters. The model contained the most frequent CAUTI species, namely *E. coli*, *P. aeruginosa*, *Klebsiella oxytoca* and *P. mirabilis*. The co-localization and relative abundance of the four species were influenced by the specificity of strains and could be manipulated by adjusting the inoculation ratio. Interspecies interactions in the model biofilm suggested that *P. aeruginosa* and *P. mirabilis* were dominating over *E. coli* and *K. oxytoca*. Although the growth of *E. coli* and *K. oxytoca* were inhibited by *P. mirabilis*, they were observed to gain enhanced tolerance to ciprofloxacin (CIP) when *P. mirabilis* was present in the biofilm. In general, species were found to have a higher tolerance to CIP in biofilms with higher numbers of species involved. This emphasizes the importance of using multispecies biofilm models to evaluate therapeutic strategies against CAUTI.

## 2. Materials and Methods

### 2.1. Bacteria and Culture Conditions

Clinical isolates used in this study originate from a collection of clinical isolates collected during routine screening at the university hospital in Umeå (Norrlands universitetssjukhus, NUS). The anonymized isolates were collected from CAUTI patients from the hospital during the period 2017–2019 and identified using matrix-assisted laser desorption/ionization–time-of-flight mass spectrometry (MALDI-TOF, Bruker, Germany) as part of hospital routine. The identity of the isolates used in this work were confirmed as *E. coli*, *P. aeruginosa*, *K. oxytoca* and *P. mirabilis* by 16s rRNA sequencing and bacterial whole genome sequencing as specified below. Isolates were stored at −80 °C in Luria–Bertani (LB) medium supplemented with 15% glycerol. For bacterial and biofilm growth, stored isolates were streaked on LB agar plates and cultured at 37 °C overnight. Single colonies were picked to inoculate 5 mL LB medium and cultured at 250 rpm at 37 °C overnight.

### 2.2. 16S rRNA Amplicon and Whole Genome Sequencing of the Clinical Isolates for Identification

Full 16S rRNA genes of the isolates were amplified by colony PCR using primers 27F (5′-AGAGTTTGATCCTGGCTCAG-3′) and 1492R (5′-GGTTACCTTGTTACGACTT-3′) for strain identification. Briefly, single pure colonies were picked from LB agar plates, suspended in 50 µL deionized H_2_O (dH_2_O, Sigma Aldrich, St. Louis, MO, USA) and boiled for 5 min. We added 1 μL boiled solutions into PCR buffer containing 10 μL PCR Mastermix (Roche, San Francisco, CA, USA), 0.8 + 0.8 μL forward and reverse primers and 7.4 μL dH_2_O, and amplified 30 cycles with an annealing temperature of 55 °C. Subsequently, PCR products were purified using the QIAquick PCR Purification Kit (QIAGEN, Germantown, MD, USA) and sent for sequencing at Eurofins Genomics (Copenhagen, Denmark). Identification was performed by blasting the achieved 16S rRNA gene sequences of the isolates in the NCBI database and selecting the strain with the highest sequence match [30].

For bacterial whole genome sequencing, DNA from an overnight culture of each isolate was extracted and purified using NucleoSpin Soil (Macherey-Nagel) and QIAquick PCR Purification Kit (QIAGEN), respectively. DNA sequencing was performed by NovaSeq PE150 (Novogene, Milton, Cambridge, UK). The short-read sequences were assembled using de novo-assembly, QIAGEN CLC Genomics Workbench. Three random assembled fragments (length over 10 kbp) were blasted in the NCBI database and the most closely related strain with the highest identification match was selected.

### 2.3. Strain Characterization

Antibiotic resistance gene (ARG) annotation was performed on the genomes of the selected strains using ARG-ANNOT [31]. The antimicrobial susceptibility of the selected strains to several antibiotics was examined by spreading 100 μL overnight bacterial culture on LB agar plates containing various concentrations of antibiotics (specified below) and cultured overnight at 37 °C. Strains were determined as sensitive to a certain concentration of antibiotics if no colonies appeared on the corresponding agar plate. The tested antibiotics and concentrations included gentamicin (GEN, 10, 30, 60 μg/mL), kanamycin (KAN, 25, 50, 100 μg/mL), tetracycline (TET, 10, 30, 60 μg/mL), ampicillin (AMP, 100, 300 μg/mL), CIP (4, 10, 20 μg/mL), trimethoprim (TMP, 20, 40, 80 μg/mL) and chloramphenicol (CHL, 10 μg/mL).

### 2.4. Biofilm Growth on Urinary Catheter Shafts 

Indwelling silicone Foley catheters (14 Fr) were provided by Biomodics ApS (Copenhagen, Denmark). The catheter shafts were manually cut into 3 mm-long, half-ring-shaped pieces (see Figure 1e). The catheter pieces were washed, autoclaved and dried before use. Artificial urine medium (AUM) [32] was used for biofilm growth. AUM was adjusted to pH 6.5 and sterilized by filtration through a 0.2 μm filter before use.

Overnight cultures of isolates were washed twice in PBS by centrifuging at 8500 rpm for 5 min and resuspended in PBS. The suspensions were all adjusted to OD = 0.1 by further dilution in PBS for inoculation on catheter pieces. For preparing monospecies biofilms, 0.5 mL bacterial suspension was added to a 1.5 mL Eppendorf tube containing a catheter piece and incubated at 37 °C at static conditions (no shaking) for 4 h for bacterial adhesion. For preparing multispecies biofilms, equal volumes of suspension from each species were mixed to a total volume of 0.5 mL. The same total suspension volume also applied for the four-species model biofilm, but the ratio of each species was adjusted as stated in the results section. Then, each catheter piece was washed twice by dipping into 1.5 mL PBS in 24-well microtiter plates using a pair of tweezers, and transferred into 0.5 mL fresh AUM in a new 1.5 mL tube for 20 h static culture at 37 °C.

### 2.5. Quantification of Biofilms by Colony Forming Unit (CFU) Enumeration

The catheter pieces and associated biofilms were washed 3× in PBS to remove the planktonic bacteria and then resuspended in 0.5 mL PBS by using FastPrep Homogenizer (MP Biomedicals, Irvine, CA, USA) twice at 4 m/s for 20 s with 5 min intervals. The resuspended biofilms were 10-fold serial diluted and 100 μL from each dilution was spread on an LB agar plate for overnight growth at 37 °C. Each species was identified and enumerated based on their distinct colony morphology (Figure 1a–d).

### 2.6. Crystal Violet Assay

The biomass of the 20 h biofilms formed on the catheter pieces was quantified by the crystal violet method also used in Ren et al. [33] with slight modification. Briefly, the catheter pieces with associated biofilms were washed 3× in PBS and transferred to 24-well microtiter plates containing 400 μL filtered 1% (*w*/*v*) crystal violet solution to stain for 30 min at room temperature. Then, the catheter pieces were washed 3× in PBS to remove excess crystal violet, transferred to 1 mL 96% vol ethanol and shaken at 300 rpm for 30 min. The ethanol solutions with dissolved crystal violet from each sample were diluted 10× and the absorbance at 590 nm was recorded by the EL 340 BioKinetics reader (BioTek Instruments, Winooski, VT, USA).

### 2.7. CIP Treatment on Planktonic Bacteria and Biofilms

CIP (Sigma Aldrich, St. Louis, MO, USA) was dissolved in 0.5 M NaOH to make a 10 mg/mL stock solution. The sensitivity of the selected isolates to CIP was measured by the minimum inhibitory concentration (MIC) test. Briefly, the overnight cultures of each isolate were adjusted to OD = 0.1 followed by 1000-fold dilution in LB broth to reach a final concentration of ~10^5^ CFUs/mL. We added 6 × 200 μL of each diluted suspension into each well of the 96 well microtiter plate and we also added different aliquots of CIP stock solution for each isolate to reach final CIP concentrations of 0, 0.16, 0.8, 4, 20, 100 μg/mL. Then, the microtiter plate was incubated at 37 °C for 24 h and OD 600 readings were performed in the EL 340 BioKinetics reader (BioTek Instruments, Winooski, VT, USA). The experiment was performed in three biological replicates.

Mono- or multispecies biofilms for both antibiotic treatment and controls (untreated biofilms) were grown for 18 h on catheter pieces (method as stated above), followed by 3× wash in PBS to remove planktonic bacteria. Then, the catheter pieces were transferred into LB broth with 100 μg/mL CIP for 6 h incubation at 37 °C, while the untreated control group was treated similarly but without adding CIP. Subsequently, the catheter pieces from both groups were washed 3× in PBS, followed by CFU quantification as stated above. The percentage strain survival was calculated as such
(1)$$\mathrm{Survival}\%\;=\frac{\mathrm{Strain}\;\mathrm{CFU}\;\mathrm{in}\;\mathrm{CIP}\;\mathrm{treated}\;\mathrm{biofilm}}{\mathrm{Strain}\;\mathrm{CFU}\;\mathrm{in}\;\mathrm{corresponding}\;\mathrm{biofilm}\;\mathrm{from}\;\mathrm{untreated}\;\mathrm{group}}\times 100\%$$

### 2.8. Statistics

Statistical analysis was performed in Python 3.7 using the following procedure. The equality of variance between the sample groups was first assessed by Levene’s test. If sample groups had equal variance, one-way ANOVA was performed followed by the Tukey post hoc test; if the variances were unequal, the Kruskal–Wallis test was performed followed by Dann’s post hoc test. Significance was accepted at *p* < 0.05. All experiments had three biological replicates.

## 3. Results

### 3.1. Identification of Clinical Isolates

A total of 34 strains from 15 indwelling catheters were included in the study. Table 1 shows the closest relative strains of the isolates according to the NCBI blasting of the 16S rRNA gene sequences. Reference numbers (CNUSM#, Clinical Isolates from NUS Multispecies) were given to each strain. Among the 34 isolates, *P. aeruginosa* (47%) was the most frequently identified strain, followed by *Escherichia fergusonii* (18%), *E. coli* (6%) and *K. oxytoca* (6%). Among the collected isolates, 91% were Gram-negative strains and only three strains were identified as *Enterococcus* or *Aerococcus*. Out of the 15 selected catheters, 12 were colonized by 2 species, catheters #5 and #9 were colonized by 3 species, while 2 species but 4 different strains were isolated from catheter #3. Since the purpose of this study was to construct a model of multispecies consortium, isolates were chosen from catheters that showed colonization by more than one bacterial species. Thus, we wish to stress that Table 1 does not correspond to the strain diversity observed in the full isolate collection and may also not represent a cross-section of all species colonizing the collected catheters.

### 3.2. Validation of Biofilm Detachment Protocol

Biofilms grown on catheter pieces were detached and resuspended in PBS using the FastPrep Homogenizer for subsequent CFU enumeration. We first validated the biofilm detachment efficiency and bacterial viability through this process. Approximately 80% of the biofilm was detached and resuspended in this step, evaluated by comparing the biomass on the homogenizer-treated and non-treated catheter pieces using crystal violet staining and OD590 absorbance readings. Bacterial viability after this treatment was evaluated by comparing the CFU of a non-treated bacterial suspension (OD = 0.1) and a suspension aliquot treated using this method together with a blank catheter piece in a 1.5 mL tube. This resulted in comparable CFU counts for the treated and non-treated suspensions.

### 3.3. Strain Selection for Four-Species Biofilm Model Establishment

In order to achieve high clinical relevance for our multispecies biofilm model, we defined three fundamental criteria for the strain selection:*P. mirabilis* must be included;The selected species should be the most common ones among all CAUTI cases;The selected strains should be able to adhere to and colonize the surface of the catheter shaft and form a stable biofilm.

The consortia from catheters #5 and #9 had a very similar strain composition and they both fulfilled criteria 2. Therefore, we based the strain selection on mixing these two consortia with the *P. mirabilis* (CNUSM025) from catheter #11 and quantified the 20 h biofilm by CFU. The strain combination was selected only if a countable number (< 200) of colonies of all involved strains could be found in one agar plate, indicating the presence of each strain within two magnitude number differences (criteria 3). If the combination proved non-successful, the strain which failed to appear on the countable plate would be replaced by the same species from another catheter for a new combination test. Following this workflow, we finally selected the four-species combination consisting of *E. coli* from catheter #12 (CNUSM028), *P. aeruginosa* from catheter #4 (CNUSM009), *K. oxytoca* from catheter #5 (CNUSM012) and *P. mirabilis* from catheter #11 (CNUSM025). The four selected strains were further identified by whole genome sequencing and confirmed to be the closest relative to *E. coli* EC28, *P. aeruginosa* PABCH14, *K. oxytoca* FDAARGOS1332 and *P. mirabilis* BB2000.

During the strain selection process, we noticed that the consortia isolated from the same catheter could prove an unsuccessful combination under the growth conditions in this study. For example, the three strains from catheter #9 were all successfully included in dual-species combinations but failed in three-species combinations. This indicates that the coexistence of species in the 20 h biofilm was more strain-specific when more species were involved.

### 3.4. Strain Characterization

ARG-ANNOT was used to perform ARG annotation on genomes of the four selected strains and the results are listed in Table 2. The susceptibility of the four strains was examined on agar plates containing the following antibiotics: GEN, KAN, TET, AMP, CIP, TMP and CHL. Among all tested antibiotics, *E. coli* (CNUSM028) was only resistant to TMP; *P. aeruginosa* (CNUSM009) was resistant to KAN, TET (up to 10 μg/mL), TMP and CHL; *K. oxytoca* (CNUSM012) was resistant to AMP and TMP; *P. mirabilis* (CNUSM025) was resistant to GEN (up to 30 μg/mL), KAN (up to 25 μg/mL), TET (up to 30 μg/mL), AMP (up to 100 μg/mL), TMP and CHL. The antibiotic sensitivity profile of the four strains partially matched the ARG annotation. For example, *apH* in *P. aeruginosa* is responsible for KAN resistance [34]; *CTX-M* in *K. oxytoca* is responsible for AMP resistance [35]; *catA* and *catB* in *P. mirabilis* and *P. aeruginosa* are responsible for CHL resistance [34]. However, some mismatches also exist between the sensitivity profiles and the ARG annotation, such as the TMP resistance of all four strains as well as a wide range of antibiotic resistance for *P. mirabilis* at relatively low antibiotic concentrations.

### 3.5. Strain Interaction and Biofilm Model Optimization

Although the selected strains were able to form a four-species biofilm on catheter shafts when incubated for 20 h, the CFUs of *E. coli* and *K. oxytoca* were consistently lower than those of the other two strains. Given the fact that these four strains had similar growth rates in liquid media when cultivated separately (data not shown), we assumed the differences in abundance in the four-species biofilm were due to synergistic/antagonistic strain interactions. Therefore, we compared the CFU of each strain in a monospecies biofilm, or in dual-, three-, four-species biofilms with the other three strains in all possible combinations (Figure 2). The medium pH of the biofilm cultures among all combinations at 20 h were approximately 7.5 and 9.0 in the absence and presence of *P. mirabilis*, respectively. As expected, *E. coli* and *K. oxytoca*, the two minorities in the four-species biofilm, showed significantly reduced numbers when they were grown in combination with *P. aeruginosa* and *P. mirabilis*. Specifically, the growth of *E. coli* was not affected by *K. oxytoca*, but it was inhibited whenever *P. aeruginosa*, or *P. mirabilis*, or both were introduced. Interestingly, the inhibitory effect of *P. mirabilis* was enhanced when *K. oxytoca* was also present, despite the fact that *K. oxytoca* did not inhibit *E. coli* in the dual species combination. The growth of *K. oxytoca* was only inhibited when *P. mirabilis* was introduced, while the growth of *P. aeruginosa* and *P. mirabilis* were not affected by introduction of any other strains.

In order to increase the abundance of *E. coli* and *K. oxytoca* in the 20 h four-species biofilms to achieve approximately equal numbers for each strain in the model, we inoculated lower amounts of the dominating strains *P. aeruginosa* and *P. mirabilis* for the weak strains to grow better. After a round of preliminary screening of ratios, we selected three promising ratios for the CFU and biomass measurements in biological triplicates (Figure 3). The CFU results indicated that the ratio of *E. coli*:*P. aeruginosa*:*K. oxytoca*:*P. mirabilis* in Mix2 (1000:1:100:10) was the best, followed by Mix3 (1000:1:100:100) and Mix1 (100:1:100:1). As expected, we noticed a proportional relation between the initial prevalence in the inoculation mixture and the CFUs observed after 20 h of biofilm growth. For example, the lower inoculation amount of *P. aeruginosa* and *K. oxytoca* in Mix2 compared to Mix1 resulted in a reduced number of CFUs of these two strains in the 20 h biofilm, while a higher inoculation amount of *P. mirabilis* in Mix3 compared to Mix2 resulted in an increase in its final CFUs. However, the unpredicted CFU changes of strains with an unchanged inoculation amount were also observed, e.g., the increased CFUs of *P. mirabilis* in Mix2 compared to Mix1, and the reduced CFUs of *E. coli* and *P. aeruginosa* in Mix3 compared to Mix2. These alterations in relative species abundances were also attributed to interspecies interactions. The biomass measurement by crystal violet indicated that *P. mirabilis* is a better biofilm matrix producer (OD590/CFU = 8.6 × 10^−7^) compared to *E. coli* (3.2 × 10^−8^), *P. aeruginosa* (5.5 × 10^−8^) and *K. oxytoca* (7.4 × 10^−8^), while the biofilm from the preferred ratio Mix2 showed an intermediate biomass production. However, it should be noted that as these interactions are strain specific (as described in Section 3.3), further studies are needed to clarify the extent to which these observations represent normal distribution for species’ interactions among CAUTI isolates.

### 3.6. CIP Susceptibility of the Strains in Multispecies Biofilms

The natural susceptibility of the model strains to CIP was first determined for each species growing planktonically under the minimum CIP concentration where no strain growth was observed (MIC). The MICs for *E. coli*, *P. aeruginosa* and *K. oxytoca* were <0.16 μg/mL, while for *P. mirabilis,* the MIC was 0.8 μg/mL. In relation to the break points from Nordicast.org, the former three strains were not resistant. However, the *P. mirabilis* strain had an MIC that was higher than the resistance breakpoint at 0.5 mg/L for Enterobacterales. As the antibiotic tolerance of the UTI strains in biofilms state are expected to be 10–100 times higher than their planktonic state [21,36], we decided to use 100 μg/mL CIP to test the survival of the model strains in mono- and multispecies biofilms (Figure 4). The monospecies biofilms of model strains were almost completely killed by 6 h CIP treatment. However, their survival rate generally increased when including more species in the polymicrobial biofilm community (Figure 4e). The survival rates of *E. coli* and *K. oxytoca* were higher in most cases when more strains were co-cultivated, while *P. aeruginosa*, the most CIP susceptible one among the model strains, only showed survival in the four-species biofilm and only at very low percentages; however, this difference was not statistically significant. This implies that a stronger protection mechanism for UTI strains may arise from a more complex community composition. In addition, *E. coli* and *K. oxytoca* showed higher survival rates when *P. mirabilis* was present in the biofilm, which is intriguing as they were both inhibited by *P. mirabilis* in the absence of antibiotics (Figure 2). Similar increased survival percentages were observed for *P. mirabilis* when co-cultivated with both *E. coli* and *K. oxytoca*, or either one of them. This mutual benefit on CIP resistance indicates a synergetic interaction between the model strains in biofilms in the presence of CIP.

## 4. Discussion

The epidemiology of CAUTI in patients that are long-term catheterized is well established in the literature [22,37] and typically involves multiple strains [16,38]. This is especially observed in immunocompromised patients experiencing long-term catheterization, or having other fundamental diseases such as diabetes [39]. Most of the clinical isolates in our collection from co-colonized catheters were identified as common CAUTI strains except *E. fergusonii*. This strain is commonly involved in cystitis and many isolates have previously been described to be multidrug resistant [40,41].

*E. coli* is the most frequent CAUTI pathogen, especially in the asymptomatic bacteriuria or infections in short-term catheterized patients where a limited number of strains are involved [14]. This might indicate that *E. coli* is a relatively weak colonizer and weak competitor compared to other UTI species. The interaction between our four selected strains showed a reduced abundance of *E. coli* in combination with both *P. aeruginosa* and/or *P. mirabilis*. Similar results were reported by Cerqueira et al. [24], showing that *E. coli* was outcompeted by *P. aeruginosa* in a dual-species biofilm. However, clinical evidence shows that *E. coli* is also common in complicated UTIs and in cases where polymicrobial co-localization is observed [10]. *E. coli* isolates from polymicrobial samples were shown to be more invasive to T24 human epithelial cells [18]. Furthermore, *E. coli* and *P. mirabilis* have been shown to use different metabolic pathways when co-existing *in vivo* and the co-inoculation of both species resulted in increased abundance and the enhanced persistence of both species in the bladders and kidneys of mice models [42]. This *E. coli–P. mirabilis* synergistic effect, reported in the literature, is different from our observation. This may be due to several reasons. One is the difference between *in vitro* and *in vivo* experimental conditions, where the surrounding environment (e.g., pH, nutrition conditions, presence of immune system) of the tested *E. coli* could be very different. A second is that the co-isolation of two species from the same catheter or urine sample does not necessarily indicate their co-localization or formation of a dual-species biofilm *in vivo*. They could colonize different sites of the catheter and affect each other remotely. In fact, evidence for the existence of inter-mixed-species biofilm is scarce due to difficulties in sampling and visualization [43]. However, the *in vivo* validation of species intermixing during catheter-associated infections can be provided by biofilm visualization based on fluorescence *in situ* hybridization (FISH) combined with CLSM (confocal laser scanning microscopy), which also allows for the analysis of species spatial organization [44]. A third explanation for the reported differences in interspecific interactions is that some interactions may be highly strain specific, as high strain diversity is observed within isolates of the same species. For example, the different strains of uropathogenic *E. coli* were shown to be very different with respect to the virulence and susceptibility to antimicrobials [45]. These differences could affect how *E. coli* interacts with other species during co-localization. 

Contrary to the variable response reported for *E. coli* in a mixed-species community, there is a consensus that *P. mirabilis* generally dominates other UTI species. Similar to our observation, *P. mirabilis* has previously been reported to inhibit the biofilm formation of *K. pneumonia* [46]. By increasing urine pH, *P. mirabilis* outcompeted other non-urease producers, such as *E. coli*, *P. aeruginosa*, and *Klebsiella* spp. during co-colonization [27,47]. However, *P. mirabilis* has also been shown to benefit from the presence of *E. coli*, *P. aeruginosa*, *K. pneumoniae* and *E. faecalis* achieving an enhanced urease activity [48]. Due to its dominance, *P. mirabilis* inhibited and outcompeted the early colonizers and rebuilt a suitable niche for itself, which explains its frequent existence in long-term catheterization even though it is not a common initial colonizer [14]. However, it has been reported that another frequent species in long-term catheterization and also a urease producer, *Morganella morganii*, could inhibit the urease activity of *P. mirabilis* and reduce the infection severity *in vivo* [29].

After establishing our model consortium, we exposed our four-species biofilm model to ciprofloxacin (CIP) and compared the antibiotic tolerance of strains in our mono- and multispecies biofilms. CIP is one of the most frequently used antibiotics in clinical CAUTI cases and one of the first choices for CAUTI according to the antibiotic usage guidelines from National Institute for Health and Care Excellence (NICE, UK). CIP is used due to its activity against a broad range of UTI pathogens, the high pathogen susceptibility to this drug, as well as its high clinical success rate [49,50]. In general, common CAUTI pathogens, including *E. coli*, *E. faecalis*, *Klebsiella* spp., *P. aeruginosa*, *P. mirabilis* and *S. aureus*, were all reported to show enhanced antibiotic tolerance in mixed-species biofilms compared to those in monospecies biofilm or planktonic states [18,21,27]. Our results showed that this enhanced tolerance was more prominent when more species were involved in the community (Figure 4e). Particularly, we observed a markedly enhanced CIP tolerance by *E. coli* in the four species biofilm setting, by far exceeding individual protective effects of the individual species on *E. coli* in dual species biofilms (Figure 4a). This indicates that community intrinsic properties are at play under these conditions and emphasizes the relevance of assessing antibiotic tolerance in a multispecies biofilm model [51]. Possible mechanisms for this community effect include antibiotic inactivation by secreted β-lactamase [17], enhanced antibiotic tolerance due to metabolic cross feeding [26], quorum sensing [52], or the induced production of biofilm matrix components such as polysaccharide β-1,3-glucan [53].

We observed mutual benefits between pairs of *P. mirabilis*–*E. coli* and *P. mirabilis*–*K. oxytoca,* where both species in each pair gained enhanced CIP tolerance when co-cultivated. This was interesting as both *E. coli* and *K. oxytoca* were inhibited in dual-species biofilms with *P. mirabilis* in the absence of CIP. This suggests that interspecies interactions change in response to environmental stress, which was also previously reported [54]. Similarly to our observation, *P. mirabilis* was reported to protect *E. coli*, *P. aeruginosa*, *Klebsiella* spp. and *Enterococcus* spp. and result in up to 4-fold enhanced tolerance against trimethoprim–sulfamethoxazole and nitrofurantoin in dual-species biofilm [27]. This enhanced tolerance was hypothesized to be attributed to an inhibited antibiotic activity induced by a high medium pH created by *P. mirabilis* [54]. This could also partly explain our observation of the enhanced CIP tolerance of *E. coli* and *K. oxytoca* in the presence of *P. mirabilis*, as there was a remarkable pH difference between the biofilm cultures in which *P. mirabilis* was involved (pH~9.0) and non-involved (pH~7.5). However, in the *P. mirabilis*–*K. oxytoca* colocalization pair, the tolerance enhancement for *P. mirabilis* was less than *K. oxytoca*, indicating that CIP inactivation might not be the only reason for the observed changes in antibiotic tolerance. A similar observation was previously reported for a *P. mirabilis–E. faecalis* pair wherein *P. mirabilis* gained a more prominent antibiotic tolerance against ceftriaxone and trimethoprim compared to *E. faecalis* [21]. This unbalanced antibiotic tolerance enhancement in a biofilm might be attributed to the spatial organization of the coexisting species where one localizes closer to the biofilm surface and thus has higher exposure to antibiotics than the other.

Most of the previous studies on the antibiotic resistance and/or tolerance of mixed-species communities were performed on the various combinations of two species. Here, we show the marked variation of antibiotic tolerance in more complex biofilms with up to four species involved. We observed that the CIP tolerance of strains was enhanced when more species were present, even though, in some cases, pairwise interactions were inhibitory in the absence of antibiotic exposure. For example, both *E. coli* and *P. aerugionsa* showed the highest survival percentage in four-species biofilms. Additionally, *K. oxytoca* and *P. mirabilis* were protected in four-species biofilms compared to monospecies, but here, other combinations were similarly or more protective. Our results indicate that, in addition to the possible pH-induced antibiotic inactivation previously suggested [54], multiple mechanisms of species interaction-induced enhanced antibiotic tolerance could be involved in a multispecies biofilm. In our four‐species biofilm, wherein three out of four selected strains are susceptible to CIP (all except *P. mirabilis*), complementary studies are needed to further clarify their synergistic effects leading to enhanced CIP tolerance. For example, the metabolite profiles of the biofilms under different conditions may explain how the different bacterial species interact.

Gram-positive pathogens, although not included in the four-species model in this study, are also frequently found in CAUTI cases, especially *Enterococcus* spp., and represent approximately 10–20% among all CAUTI isolated microorganisms [14,38,55]. *Enterococcus* spp. were reported to have various synergistic and antagonistic interactions with other common CAUTI pathogens, as discussed above. Therefore, it will be interesting in the future to include *Enterococcus* spp. in multispecies model communities to further investigate species interactions and the subsequent antimicrobial tolerance.

## 5. Conclusions

This study presents an *in vitro* four-species biofilm model using the clinical isolates of the most frequent CAUTI pathogens *E. coli*, *P. aeruginosa*, *K. oxytoca* and *P. mirabilis* to study species interactions and the inhibitory effect of ciprofloxacin on mono- and multispecies biofilm settings. The relative abundance of each species in the biofilm was highly dependent on the inoculation ratio, interspecies interactions and strain specificity. *P. mirabilis* was observed to outcompete *E. coli* and *K. oxytoca* in the absence of antibiotics, whereas increased synergistic interactions and protection was observed upon CIP treatment. In general, individual species survival, when exposed to CIP, increased with the increasing number of species present in the biofilm. This indicates the necessity to perform testing using complex communities consisting of more than two species to evaluate biofilm responses to antibiotics or other anti-CAUTI strategies. The robustness of multispecies biofilm against the action of ciprofloxacin emphasizes the importance of using multispecies biofilm for research into antifouling and/or antimicrobial materials preventing the colonization of medical devices.

## Figures and Tables

**Figure 1 microorganisms-10-01207-f001:**
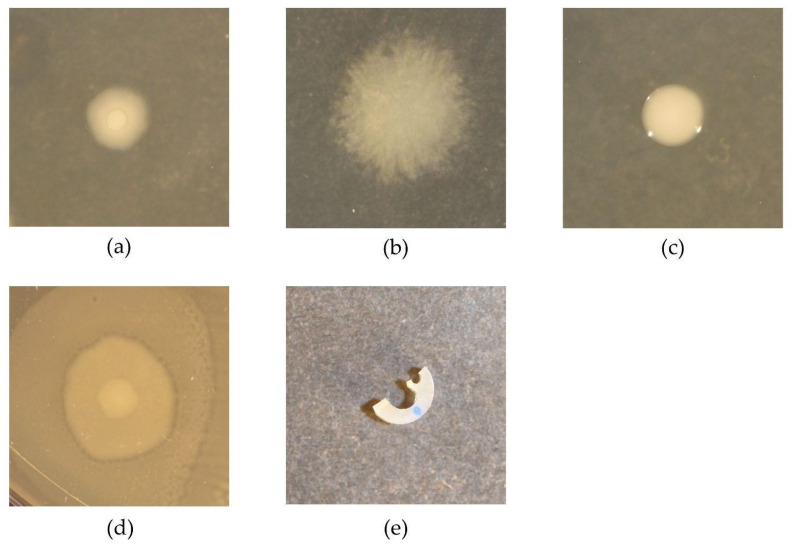
Illustration of pathogen colonies (**a**–**d**) and catheter piece (**e**). Colonies of *E. coli* (**a**), *P. aeruginosa* (**b**), *K. oxytoca* (**c**) and *P. mirabilis* (**d**) were acquired on an LB agar plate after overnight growth at 37 °C. Panel (**e**) shows a catheter piece for biofilm growth, manually cut from the shaft of a silicone Foley catheter (Biomodics ApS).

**Figure 2 microorganisms-10-01207-f002:**
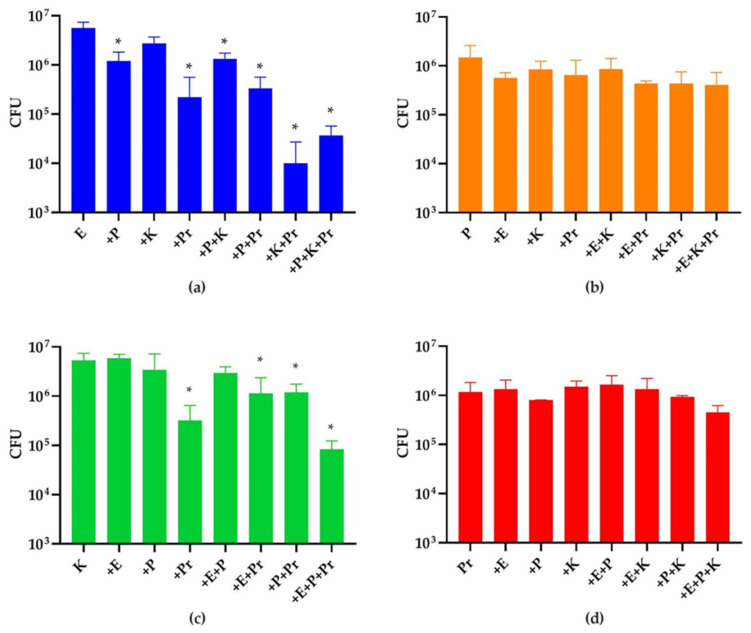
Species abundance in 20 h mono- and multispecies biofilms. CFUs of (**a**) *E. coli* (E), (**b**) *P. aeruginosa* (P), (**c**) *K. oxytoca* (K) and (**d**) *P. mirabilis* (Pr) were quantified. The first left column of each panel shows the CFUs of each species in monospecies biofilms. The rest of the columns show CFUs of each strain in multispecies biofilm of different combinations as indicated in x axis. The * indicates a significant difference (*p* < 0.05) in the strain’s CFUs in such a combination compared to that in its monospecies biofilm. The experiment was performed in three biological replicates.

**Figure 3 microorganisms-10-01207-f003:**
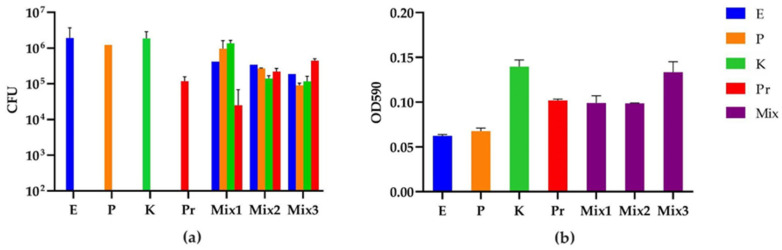
CFU and biomass of the mono- and multispecies biofilms from different strain inoculation ratios of *E. coli* (E), *P. aeruginosa* (P), *K. oxytoca* (K) and *P. mirabilis* (Pr): (**a**) CFU quantification (CFU/mL); and (**b**) absorbance of crystal violet at 590 nm indicating the biomass of each biofilm. The 20 h multispecies biofilms were grown from three inoculation strain mixtures with ratios of E:P:K:Pr for Mix1 = 100:1:100:1, Mix2 = 1000:1:100:10 and Mix3 = 1000:1:100:100. Eluted crystal violet from each biofilm was diluted 1:10 in 96% ethanol before the OD590 measurement. The background OD590 from blank 96% ethanol was 0.045. The OD590 of the negative control (crystal violet-treated bare catheter piece) was 0.009. The experiment was performed in three biological replicates.

**Figure 4 microorganisms-10-01207-f004:**
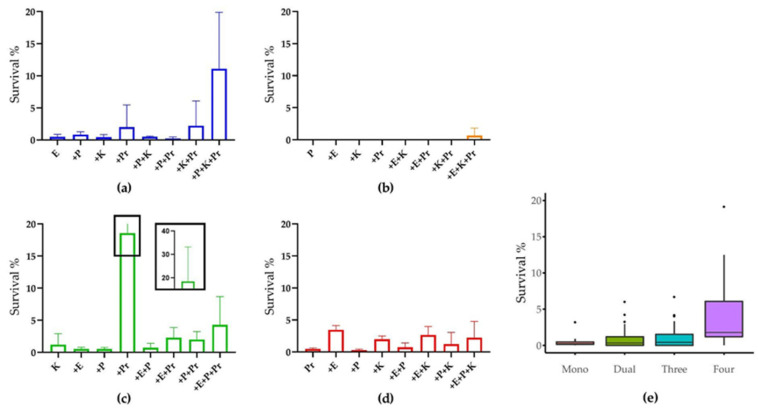
CIP susceptibility of strains in mono- and multispecies biofilms. Survival rate of (**a**) *E. coli*, (**b**) *P. aeruginosa*, (**c**) *K. oxytoca* and (**d**) *P. mirabilis* in monospecies biofilm or in combination with the other strains. (**e**) Overview of species survival rates in biofilms containing different numbers of species. The dots in penal (**e**) represent outliers within each data set. The experiment was performed in three biological replicates.

**Table 1 microorganisms-10-01207-t001:** Closest relatives of the isolates from indwelled catheters based on 16S rRNA gene sequencing.

Catheter No.	Closest Relative	Similarity (%)	Ref. No.
1	*Pseudomonas aeruginosa* DSM 50071	100	CNUSM001
	*Escherichia fergusonii* ATCC 35469	99.57	CNUSM002
2	*Pseudomonas aeruginosa* DSM 50071	99.81	CNUSM003
	*Aerococcus urinae* NBRC 15544	99.44	CNUSM004
3	*Pseudomonas aeruginosa* DSM 50071	100	CNUSM005
	*Pseudomonas aeruginosa* DSM 50071	99.89	CNUSM006
	*Enterococcus faecium* DSM 20477	99.72	CNUSM007
	*Pseudomonas aeruginosa* DSM 50071	99.46	CNUSM008
4	*Pseudomonas aeruginosa* DSM 50071	99.91	CNUSM009
	*Proteus vulgaris* ATCC 29905	99.9	CNUSM010
5	*Pseudomonas aeruginosa* DSM 50071	100	CNUSM011
	*Klebsiella oxytoca* ATCC 13182	99.64	CNUSM012
	*Escherichia fergusonii* ATCC 35469	99.42	CNUSM013
6	*Pseudomonas aeruginosa* DSM 50071	100	CNUSM014
	*Citrobacter freundii* NBRC 12681	100	CNUSM015
7	*Pseudomonas aeruginosa* DSM 50071	99.43	CNUSM016
	*Klebsiella grimontii* SB73	99.41	CNUSM017
8	*Pseudomonas aeruginosa* DSM 50071	100	CNUSM018
	*Klebsiella variicola* F2R9	99.5	CNUSM019
9	*Pseudomonas aeruginosa* DSM 50071	100	CNUSM020
	*Escherichia fergusonii* ATCC 35469	99.81	CNUSM021
	*Klebsiella oxytoca* ATCC 13182	99.66	CNUSM022
10	*Pseudomonas aeruginosa* DSM 50071	100	CNUSM023
	*Enterococcus faecalis* ATCC 19433	99.62	CNUSM024
11	*Proteus mirabilis* JCM 1669	99.72	CNUSM025
	*Escherichia coli* NBRC 102203	99.58	CNUSM026
12	*Pseudomonas aeruginosa* DSM 50071	99.82	CNUSM027
	*Escherichia coli* NBRC 102203	99.56	CNUSM028
13	*Escherichia fergusonii* ATCC 35469	99.82	CNUSM029
	*Pseudomonas aeruginosa* DSM 50071	99.57	CNUSM030
14	*Pseudomonas aeruginosa* DSM 50071	99.83	CNUSM031
	*Escherichia fergusonii* ATCC 35469	99.64	CNUSM032
15	*Pseudomonas aeruginosa* DSM 50071	99.91	CNUSM033
	*Escherichia fergusonii* ATCC 35469	99.64	CNUSM034

CNUSM: Clinical Isolates from NUS Multi‐species.

**Table 2 microorganisms-10-01207-t002:** ARG annotation of the selected strains.

Strain	ARG	Antibiotic Class
*E. coli* (CNUSM028)	*Penicillin_Binding_Protein_Ecoli*	Β-lactams
	*AmpC2*	Β-lactams
	*ZEG*	Β-lactams
	*ACT*	Β-lactams
*P. aeruginosa* (CNUSM009)	*apH*	Aminoglycosides
	*OXA*	Β-lactams
	*PDC*	Β-lactams
	*mcr*	Colistins
	*OqxBgb*	Fluoroquinolones
	*bcr1*	Phenicols
	*catB*	Phenicols
*K. oxytoca* (CNUSM012)	*OXY*	Β-lactams
	*CTX-M*	Β-lactams
	*ampH*	Β-lactams
	*OqxBgb*	Fluoroquinolones
*P. mirabilis* (CNUSM025)	*tetJ*	Tetracyclines
	*catA*	Phenicols

## Data Availability

Full genome sequences of *E. coli_*CNUSM028, *P. aeruginosa_*CNUSM009, *K. oxytoca_*CNUSM012 and *P. mirabilis_*CNUSM025 were deposited in the NCBI database: https://www.ncbi.nlm.nih.gov/genome/microbes/.

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
