# Peer review of "Enhanced Antibiotic Tolerance of an In Vitro Multispecies Uropathogen Biofilm Model, Useful for Studies of Catheter-Associated Urinary Tract Infections"

_microorganisms, 2022, doi:10.3390/microorganisms10061207_

Round 1
Reviewer 1 Report
In this study, Hou et al characterized the CAUTI caused by multi-species biofilm. They showed this structure contributed to the tolerance to antibiotic ciprofloxacin. The topic is interesting, however, I have some comments as described below,
Major comments:
- Detailed information of four strains used in this study should be offered. e.g. the antimicrobial susceptibility. In addition, it's not clear the purpose of WGS for these 4 strains? Since the authors have performed WGS, they should investigate whether these strains have toxin genes to inhibit the growth of other bacterial species.
- Fig. 3B. The biofilm production determined by CV stain was very low compared to the other studies. Therefore, the authors should describe the background OD590 results of this assay (negative control results).
- Fig. 4C. The results showed that K + Pr had higher a survival rate (compared to K only). However, the authors should explain why Pr + K in Fig. 4D did not show the same effect on bacterial survival.
- The authors can test the effect of multi-species biofilm on the tolerance to other antibiotics. In addition, this effect may be strain-specific. The authors can test other strains isolated from patients with CAUIT.
Minor comments:
- The name of bacterial species should be written in italic. e.g. lines 66, 69, 70...
- Line 85, "pneumonia" should be "pneumoniae"
- Lines 123, 125. "H2O" should be "H2O"
- Line 124, "1 µl" should be "One µl".
- Line 135, The accession numbers of genomes of 4 strains in NCBI database should be offered.
- Line 361. "et. al" should be "et al."
- Line 423. "50" should be "(50)"
- Lines 432 and 433. The names of antibiotics should not be capitalized.
Reviewer 2 Report
This is a very interesting in vitro study of CAUTI showing the enhance antibiotic tolerance of a multi-species uropathogen biofilm model. The authors could demonstrate that in their four-species in vitro biofilm model on silicone Foley catheter pieces involving clinical strains (E. coli, P. aeruginosa, K. oxytoca, P. mirabilis) by interspecies interactions antibiotic treatment, e.g. with ciprofloxacin, may become an increasing problem.
The in-vitro study is well performed and described, The clinical relevance to treat with antibiotics such CAUTI is still a big problem, because the usual routine susceptibility tests maybe misleading if multi-species biofilm CAUTI have to be treated.
One further comment: Line 38: There is no such as "Asymptomatic UTI", please use the term "asymptomatic bacteriuria", which is not a UTI.
Round 2
Reviewer 1 Report
The authors have satisfactorily addressed most of my concerns. Therefore, I recommend acceptance of this manuscript.